# Localization of Increased Noise at Operating Speed of a Passenger Wagon

**Ján Ďungel [1], Peter Zvolenský [2], Juraj Grenčík [2],\*, Lukáš Leštinský [2] and Ján Krivda [3]**

1   Department of Engineering Services, CEIT, a.s., Univerzitná 8661/6A, 010 08 Žilina, Slovakia; dungeljan@gmail.com
2   Department of Transport and Handling Machines, University of Žilina, 010 26 Žilina, Slovakia; peter.zvolensky@fstroj.uniza.sk (P.Z.); lukas.lestinsky@fstroj.uniza.sk (L.L.)
3   Department of Srategic Development, RETEX, a.s., Unádraží 894, 672 01 Moravský Krumlov, Czech Republic; krivda@retex.cz
\*   Correspondence: juraj.grencik@fstroj.uniza.sk; Tel.: +421-41-513-2553

**Abstract:** Noise generated by railway wagons in operation is produced by large numbers of noise sources. Although the railway transport is considered to be environmental friendly, especially in production of $CO_2$ emissions, noise is one of problems that should be solved to keep the railway transport competitive and sustainable in future. In the EU, there is a strong permanent legislation pressure on interior and exterior noise reduction in railway transport. In the last years in Slovakia, besides modernization of existing passenger wagons fleet as a cheaper option of transport quality improvement, quite a number of coaches have been newly manufactured, too. The new design is usually aimed at increased speed, higher travel comfort, in which reduction of noise levels is expected. However, not always the new designs meet all expectations. Noise generation and propagation is a complex system and should be treated such from the beginning. There are possibilities to simulate the structural natural frequencies to predict vibrations and sound generated by these vibrations. However, the real picture about sound fields can be obtained only by practical measurements. Simulations of the wagon's natural frequencies and mode shapes and measurements in real operation using a digital acoustic camera Soundcam have been done, which showed that for the calculated speeds the largest share of noise from the chassis was not radiated through the floor of the wagon, as was expected, but through the ceiling of the wagon. To improve the acoustic properties of the wagon at higher speed, it was proposed to use high-volume textile insulation in the ceiling of the wagon. The paper briefly presents modern research approaches in the search for ways to reduce internal noise in selected wagons used in normal operation on the Slovak railways.

**Keywords:** noise; passenger wagon; noise simulation; noise measurements; noise localization

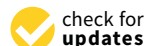



## 1. Introduction

When seeking a sustainable public transport, railway transport seems to be a good option for future development. Assessment of railway transport environmental friendliness have been studied in number of studies, e.g., [1] Perhaps the most important indicator of sustainability is quantity of greenhouse gas emissions [2]. Railway transport from its beginnings, parallel to steam traction, used electric locomotives for train's propulsion. Currently, most of major railway lines in Europe are electrified and even on non-electrified lines lot of development has been done recently with hybrid locomotives, hydrogen powered locomotives, etc. Certainly, when considering total energy consumed by railway transport, not only propulsion, yet energy used for vehicles manufacture, construction of infrastructure, traffic operation control etc. should be included [3]. Anyway, railway transport is considered as a "green" and is largely supported by various research programs, e.g., [4].

What seems to be a clear advantage of the railway transport in terms of energy demands and thus imposing lesser environmental burden compared to road or air transport, there are other parameters of the environment where railway transport is not so environmental friendly. Among them, the noise pollution is probably the most significant [5,6]. Extensive research studies on noise annoyance caused by railway traffic have been carried out around the world [7,8]. Large residential areas along railroads are significantly affected by noise emissions [9,10]. Various effects of noise can be distinguished, some of them specific, e.g., ground-borne noise form railway tunnels and its effect on sleep quality [11]. Study [12] compares the noise effects of different transport modes on sleep.

Effect or noise is primarily perceived as disturbing or annoying sound, though at higher sound pressure levels it may cause permanent damage of hearing. Mostly this is not a case of railway noise as the sound peaks are not too high. The noise levels of railway lines are ranging from 60 to 80 dB(A), depending on the flow and the limit speed of the railway. Dwellings at a distance of 70 to 950 m are under the influence of noise of railway lines exceeding permissible limits. The most effective measures to reduce noise of trains seem to be a set of measures aimed at both reducing the noise at the source and on the propagation path, i.e., noise screens and their combination with measures to improve rolling stock and acoustic properties of the track [13].

Study [14] focused on the noise caused by the train traffic and its negative impact on people living up to 100 m from the railroad. The survey measured the noise created by passenger trains in Lithuania at selected railway stations in the evening. The study considered and evaluated also the climatic conditions, train types and the number of wagons in the train. Railway ground-borne noise and vibration mitigation throughout the whole life cycle and presents results of assessment of using geosynthetics, metamaterials and ground improvement for long/term improvements are described in [15].

Besides airborne noise transmission, ground-borne vibration transmitted into buildings and perceived either as whole-body vibration or as low frequency noise, is another unfavourable effect of railway transport. It can also affect sensitive equipment but it is generally at a level that is too low to cause structural or cosmetic damage to buildings. There are numerous empirical and numerical prediction methods being developed aiming at possible mitigation methods [16,17].

Various methods and ways of noise reduction have been searched so far. Unconventional usage of conventional material is described in [18]. The study describes the development of sound-absorbing materials from production waste and natural materials that are easily decomposed in the environment. In this study, the sound-absorbing properties of nonwoven webs produced from chicken feather fibres, a by-product in chicken production and a significant amount of waste, were investigated. For this purpose, nonwoven web samples with different parameters were produced by using different binding materials by using thermal bonding method. The sound absorption coefficient and sound transmission loss values of the samples were measured and evaluated. As a result of the analyses, the influence parameters such as thickness, bulk density and porosity on the sound insulation properties of the produced samples was revealed. Studies have shown that nonwoven webs from chicken feather fibres can be used as soundproof materials because of their good sound-absorbing properties.

To design a noise reduction structure is not a simple task. For example, problems occurred with design of composite covers which protect the chassis of a modern traction vehicle moving at high speed on Polish railway routes [19]. Such covers must have appropriate strength properties and high surface resistance to external damage, while limiting the influence of the impact of elements on the cover, and the impact of external sources of noise and vibrations on the interior of the vehicle. They have a sandwich structure and are made of a polymer composite. Particular attention had to be paid to the required structure of the cover. Structural analysis is necessary for newly developed railway vehicles [20], while anticipated noise properties must be considered.

Specific acoustic properties have been studied in number of research works. A typical source of noise on rail vehicles is squeal form wheel/rail contact. The research can be dated back to second mod of twentieth century [21] up to present. The last one [22], in contrast to the previous research that did not consider longitudinal creepage to be relevant to squeal, demonstrates the importance of longitudinal creepage as a source of instability for squeal. They showed that modelling the dynamics of a wheel including the effects of a rotating wheel under moving load excitation is crucial to understanding the instability and oscillations leading to noise. The noise generated by wheels is transmitted further principally by two paths—air-borne and structure-borne. Depending on car-body structure and its structural/acoustic properties noise is perceived in the interior of a vehicle by passengers and staff.

A comprehensive overview on railway noise and vibration, its mechanisms, modelling and means of control is presented in [23]. The book brings together coverage of the theory of railway noise and vibration with practical applications of noise control technology at source to solve noise and vibration problems from railways. Each source of noise and vibration is described in a systematic way: rolling noise, curve squeal, bridge noise, aerodynamic noise, ground vibration and ground-borne noise, and vehicle interior noise.

The International Union of Railways (UIC) has defined long term goals for railway noise in a publication called "Rail Strategy 2030 and beyond for Environment, Energy and Sustainable Mobility" [24]. Nevertheless, once this is implemented, other type of rolling stock needs to be further developed for low noise emissions. The vision from [24] is that "By 2030 noise and vibration is no longer a problem for the railways—meaning actual be lowered 10–20 dB (A) compared to 2005 levels".

For example, the document "High-Speed Ground Transportation Noise and Vibration Impact Assessment" [25] set forth in the US allows to calculate and assess the predicted noise levels, including from "very high-speed train" (with a maximum speed of 400 km/h) and from 'maglev' train category representing the upper range of speed performance up to 480 kmh. There are several subsource components in this technique, including a propulsion, wheel-rail and aerodynamic subsources (train nose, wheel region, pantograph). The calculation model is based on noise generation processes research of such trains as the Amtrak Acela, French TGV (Thalys, Atlantique, Reseau, Duplex), Swedish X2000, Eurostar, Korean KTX-II, KTX-III, German IC T, ICE 3, AVE S103, Pendolino ETR450 [25].

*Simulation Methods in Noise and Vibrations*

New prospects and trends in noise reduction of railway vehicles are often based on simulation and various experimental methods. The great advantage of simulations is that it is possible to determine how a certain vehicle will behave before it goes into production and thus it is possible to speed up and improve the design process and also reduce its development costs. At present, it is possible to use a number of software packages that can be applied in simulations to strength analyzes of the structure, flow in fluid systems, noise propagation and the like. In order to be able to perform simulations in the required accuracy, it is necessary to have the most accurate model of the simulated body or object. For simulations, conditions, such as different loads, operating modes, environment, etc., are necessary for assumption how the tested object will behave in reality [26].

There are a number of types of vehicle simulation models:

- single-mass vehicle model (model with one degree of freedom) used to simulate a simple vibration response,
- dual-mass and planar vehicle model,
- spatial vehicle models—the most complex models.

In addition to considering individual vehicle parts as discrete bodies, it is possible to consider vehicle parts as flexible bodies and to move from discrete dynamics to the dynamics of continuous bodies in dealing with vehicle oscillations. In the process of creation such models, the Finite Element Method (FEM) has a dominant position among the approximate methods of continuous body mechanics.

In the noise simulations and measurement, frequency analysis has its inevitable role. The purpose of frequency analysis is to break down a complex signal into its components at various frequencies. Mathematicians and theoretical engineers tend to interpret "components" as the results of a Fourier analysis, while practical engineers often think in terms of measurements made with filters tuned to different frequencies [27].

The Fast Fourier Transform (FFT) is an algorithm or calculation procedure for obtaining the Discrete Fourier Transform (DFT) with greatly reduced number of arithmetic operations compared with a direct evaluation [28]. It permits the evaluation of a large number of functions applicable to multi-channel measurements, e.g., correlation, coherence, transfer functions, etc. [29]

There are many noise simulation software currently available. As part of solving research tasks, we used the COMSOL Multiphysics® and ANSYS software. In both cases, it was FEM-based simulation. The software tools contain modules for strength analysis, fluid flow and many other physical processes, in addition they also provide the acoustic module [30,31].

As an example of acoustic simulation, a disc brake squealing noise done by ANSYS acoustic module is presented in Figure 1.

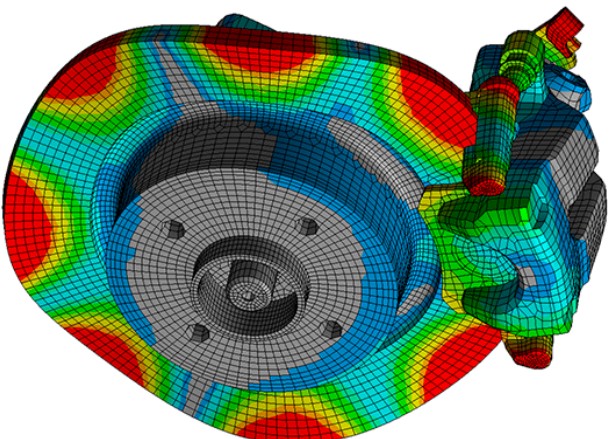

**Figure 1.** Example of simulation of disc brake squealing in the ANSYS program [32].

The design, material and technological solutions of the car body of wagons must also follow the environmental criteria as well as the comfort parameters for passengers in the different types of passenger trains. Noise and vibration are especially perceived by passengers' senses and their share in the total comfort is very significant. Therefore there are efforts to eliminate internal noise and vibration as much as possible.

The article is focused on the following areas:

- computer simulations of the generation and transmission of noise in the body of a railway passenger car
- monitoring of vibro-acoustic phenomena
- experimental research of level and frequency of internal noise
- use of an acoustic camera to display sources and paths of noise propagation and identification of critical noise emission points
- proposal for utilization of recyclable sound insulation materials for noise reduction

## 2. Localization of Noise at Wagon Operating Speed

During the running of railway wagons, a large number of noise sources are generated, such as rolling noise, braking noise, aerodynamic noise, construction vibration noise, or noise from various equipment (air conditioning, door controls, etc.).

Continuous growth of the legislation requirements on the noise reduction is pushing manufacturers, already in the development and subsequent production of passenger

railway wagons, to consider that the noise levels in the interior as well as in the exterior should be as low as possible.

When designing a structure that should meet the required parameters, it is currently possible to use a wide range of tools, thanks to which we can determine the behaviour of the wagon before the test run of the prototype was manufactured. Thanks to the use of simulations, it is possible to save significant financial costs during development and at the same time it is possible to avoid undesirable operating conditions, while designing a structure with better acoustic properties.

The same method can be applied to the existing structure, where it is possible to identify potential shortcomings and propose adequate solutions. In the study we deal with the already existing design of a passenger railway car, examining whether it is possible to achieve better acoustic comfort for passengers by applying better insulation.

Before planning the measurements, a 3D model of the car body was created, which subsequently was planned to be subject to measurements (Figure 2). We investigated the 3D model for its natural (Eigen) frequencies using the finite element method in ANSYS.

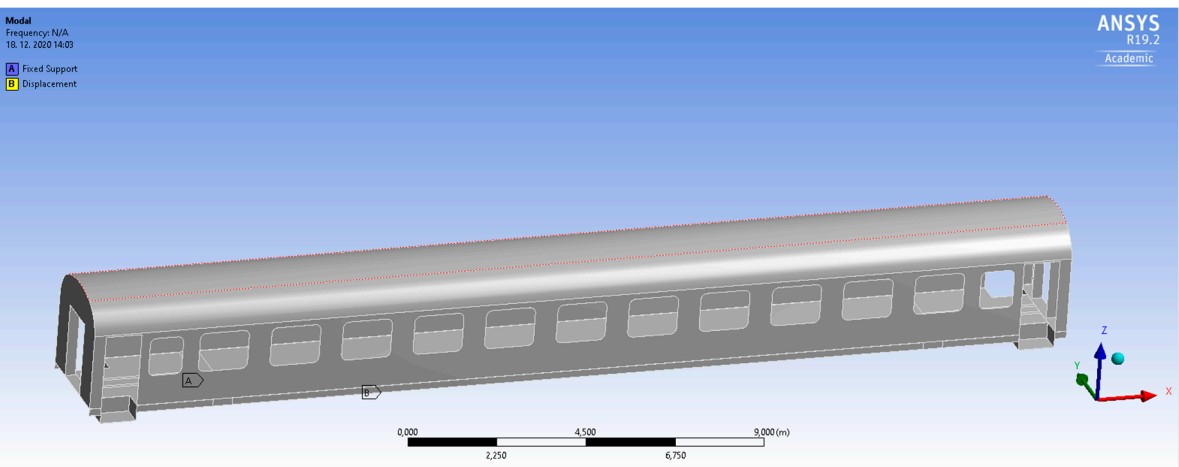

**Figure 2.** 3D model of the passenger coach car body.

### 2.1. Mechanical Resonance and Natural Frequences

The natural frequency (or Eigen frequency) of an undamped system is the frequency at which the systems tend to vibrate when excited and no damping component is present. A vibrating object or system has one or more natural frequencies, depending on whether we consider a continuous or discrete system. A continuous system has an infinite number of natural frequencies and a discrete system has a finite number of natural frequencies.

The shape of motion of a system vibrating at its natural frequency is called the mode shape. Mode shapes are intrinsic in the sense that they can move independently of each other, and thus the induction of one mode shape will never cause the movement of another. Each mode shape is characterized by one of its natural frequencies, depending on the number of degrees of freedom of the system.

The mode shape is a motion in which all parts of this system move sinusoidally with a constant phase and frequency. The frequency at which the system vibrates at its natural frequency is called the resonant frequency. Physical objects (flexible bodies) have a set of mode shapes, the resonant frequencies of which depend on their structure, material and boundary conditions.

Mechanical resonance, i.e., the conformity of the natural mechanical frequencies of the wagon structure with the frequency of the incoming pulses (oscillations, shocks) is an undesirable phenomenon, and can cause damage to the oscillating part as well as to the part to which the oscillation is transmitted. Forces excited by a dynamic load at the same frequency identical to some natural frequency of the structure can cause resonance

accompanied by an increase in the noise level of the structure. In the case of rolling stock, the dominant source of excitation is the rolling of the wheel along the rail, which is then transmitted to the body of the wagon via the bogie and the pin. During normal operation, this effect is minimal on the upgraded line, unless the frequency of rotation of the wheels at the prescribed operating speed is inherent in the shape of the wagon body.

### 2.2. Simulations of Natural Frequencies

The simulations resulted in several mode shapes, each of which was characterized by a particular frequency. During the simulation we searched for whether one of the dominant sources of noise (noise generated from the bogie when rolling the wheels on the rail at a specific speed) does not match the natural frequency of the car body. These frequencies corresponded to the frequencies of rotation of the wheels at the speeds that the wagon would reach, and it was found that vibrations occurred on the roof of the wagon.

Figure 3 shows the first mode shape, which is characterized by a frequency of 9.28 Hz. Thus, the frequency that excited it corresponds to the rotation frequency of the wheels at a speed of 96.55 km/h.

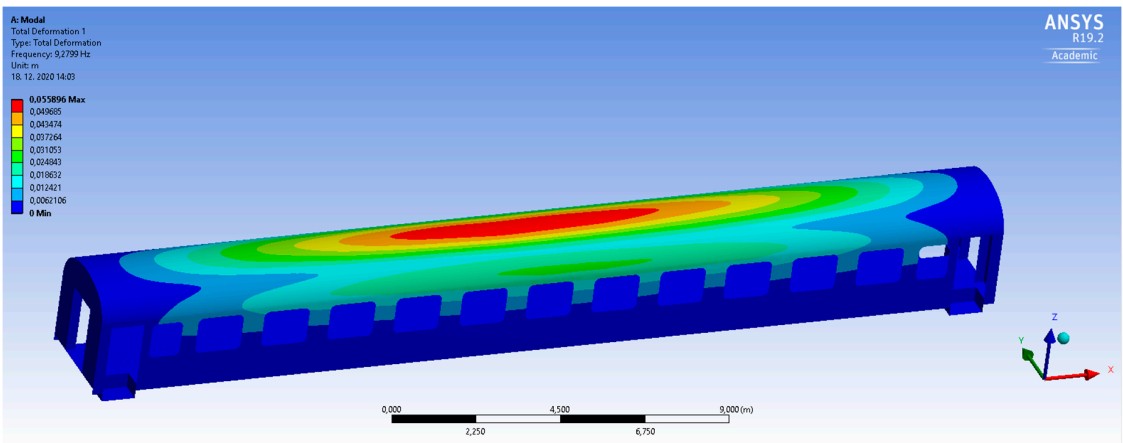

**Figure 3.** The first mode shape of the passenger coach roof.

Figure 4 shows a second natural shape whose frequency of 15.17 Hz corresponds to the frequency of rotation of the wheels at a speed of 157.9 km/h. This speed (160 km/h) is typical for upgraded railway tracks in Slovakia and most of running time is at this speed. So resonance should be avoided.

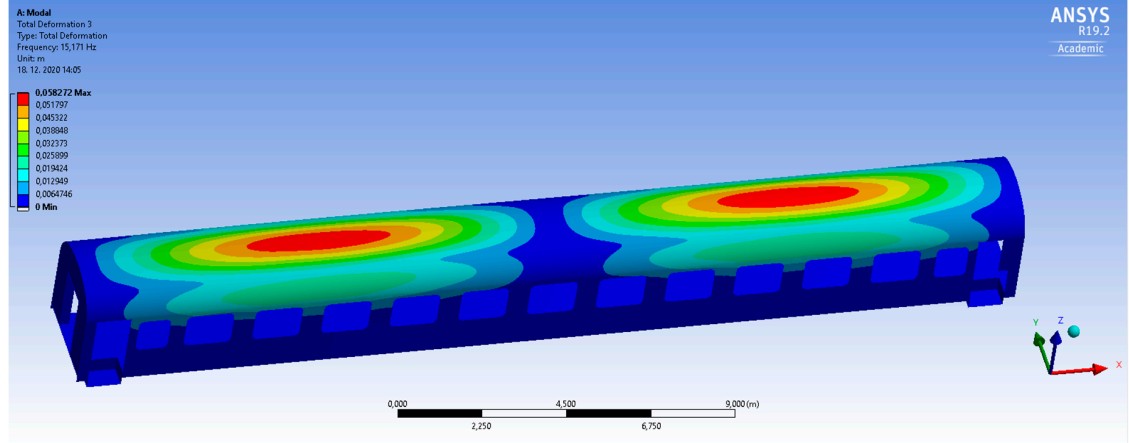

**Figure 4.** The second mode shape of the passenger coach roof.

Figure 5 shows a bottom of the car body. The structure of the underframe is very stiff (made of steel sheets thickness of 6 mm), so resonance effect of oscillations that occur on the roof do not occur in the underframe (bottom of the car-body).

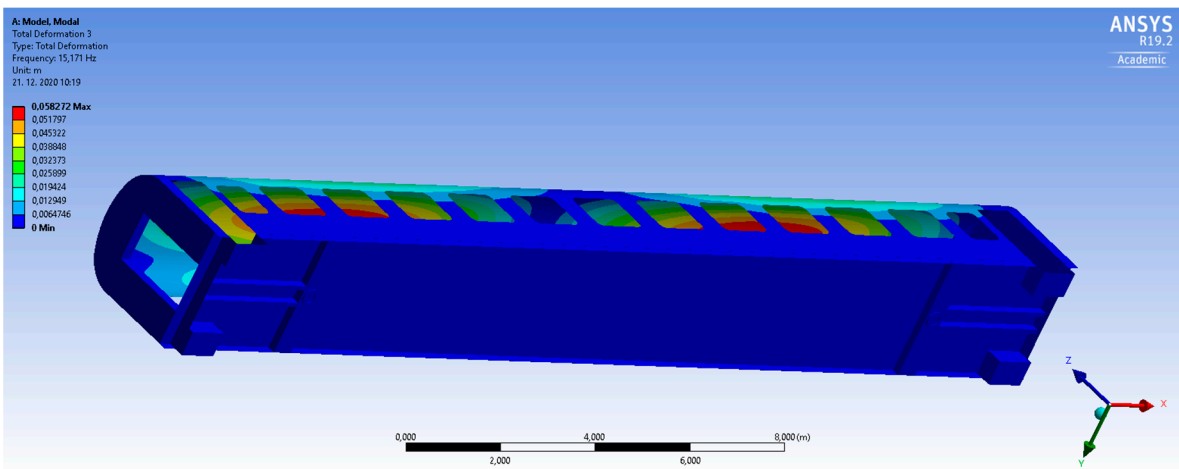

**Figure 5.** Bottom of the passenger coach car body—without mode shapes.

## 3. Results—Passenger Wagon Noise Measurements

To verify the simulation, a measurement was performed on the corridor line using a digital acoustic camera Soundcam using far field analysis (beamforming) for sound field analysis and visualization [33]. Average sound pressure level (Figure 6) visualised by SONAH (Statistically Optimized Near-field Acoustical Holography) is designed for the offline analysis and is based on the geometry of the sound source. The frequency range is 1500–9300 Hz. The key to using this algorithm is to represent the sound pressure at specific positions as a finite sum of the planes of transient waves.

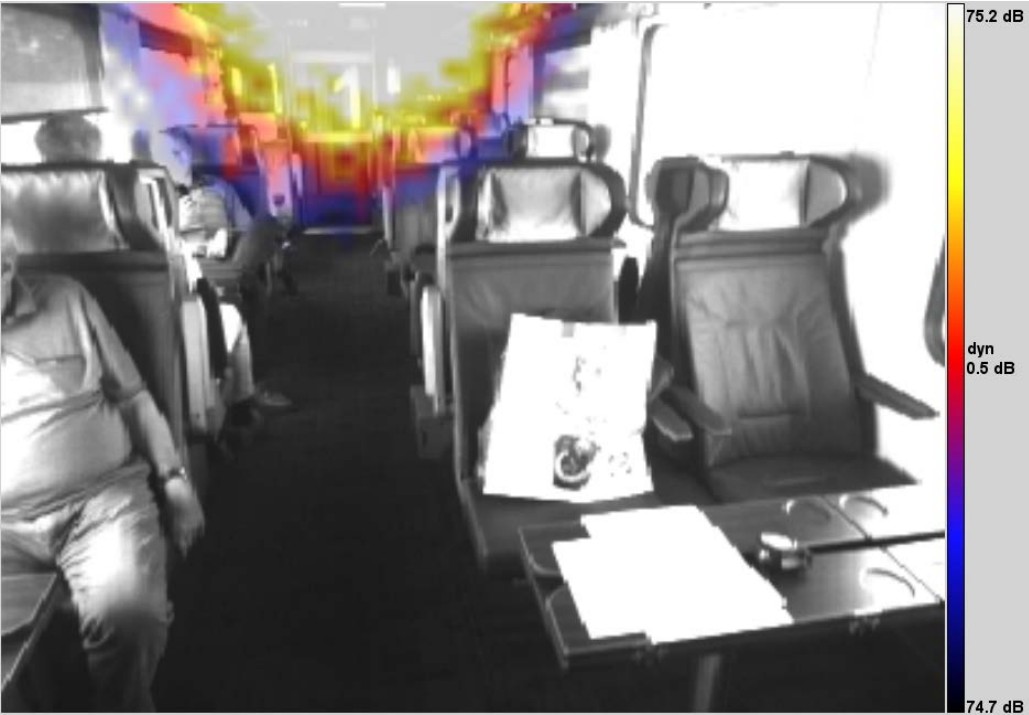

**Figure 6.** Recording from the acoustic camera—visualization of interior acoustic field.

The measurement showed that for the calculated speeds the largest share of noise from the chassis was not radiated through the floor of the wagon, but through the ceiling of the wagon. For this wagon, the application of better acoustic insulation to the chassis does not make sense, because its maximum speed of 160 km/h will not improve its acoustic comfort for passengers. If we want to improve the acoustic properties of the wagon at this speed, it is necessary to use high—volume textile insulation in the ceiling of the wagon, which would be specially applied to the ceiling and would probably change the wagon's mode shapes.

The image from the acoustic camera (Figure 6) reveals that the noise radiates from the ceiling of the wagon. The recording was made at the indicated speed of 160 km/h. Subsequent analysis of the noise frequency revealed that the noise frequencies correspond to the rotational frequencies of the wheels at a given speed, thus confirming the results of the simulation of their own shapes.

### 3.1. Measurement of the Effect of Noise Increase on the Passenger

The second part of the measurement was performed according to the standard ČSN EN ISO 3381. The measuring devices B&K 2250 and B&K Pulse 3560B were used for the measurement. The measurement was performed on a calibrated section of the corridor line. As can be seen on the noise map (Figure 7), the greatest noise is above the bogies of the wagon. However, most of this noise is emitted from the ceiling.

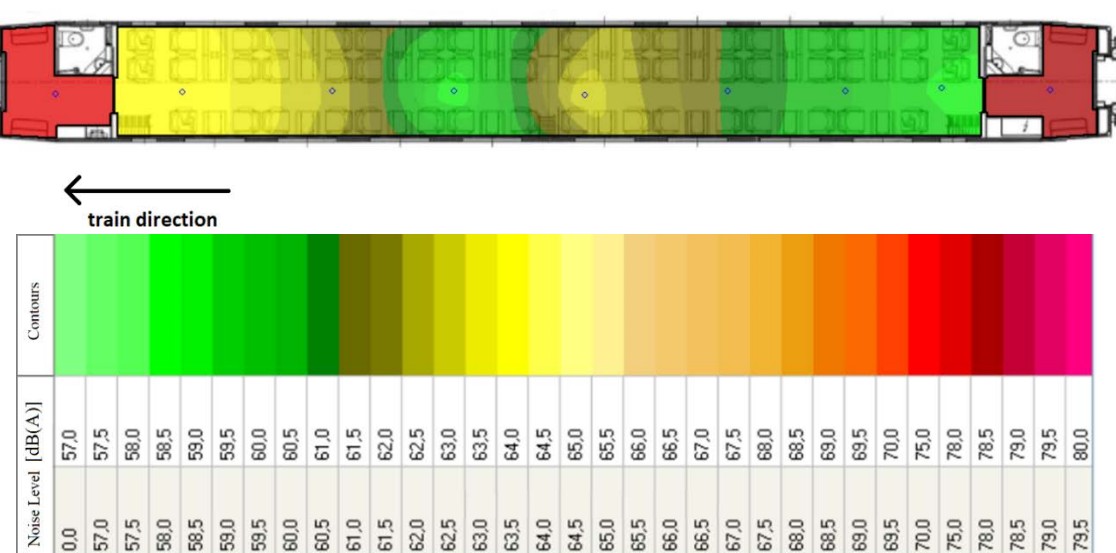

**Figure 7.** Noise map of the passenger coach.

Values of 1/3 octave frequency analysis of noise measured in interior of single-compartment passenger car (measuring positions were located above the chassis and in the middle of the car) speed 160 km/h, air conditioning switched-off, are presented in Figure 8 and Table 1:

### 3.2. Measurement of the Exterion Noise of the Passenger Wagon

Measurements of exterior noise of the passenger car have been done, too. Acoustic camera Soundcam was placed on the bridge over the railway line on the newly reconstructed track. The speed of the train was of 160 km/h. The pictures showing acoustic field are presented in Figures 9 and 10.

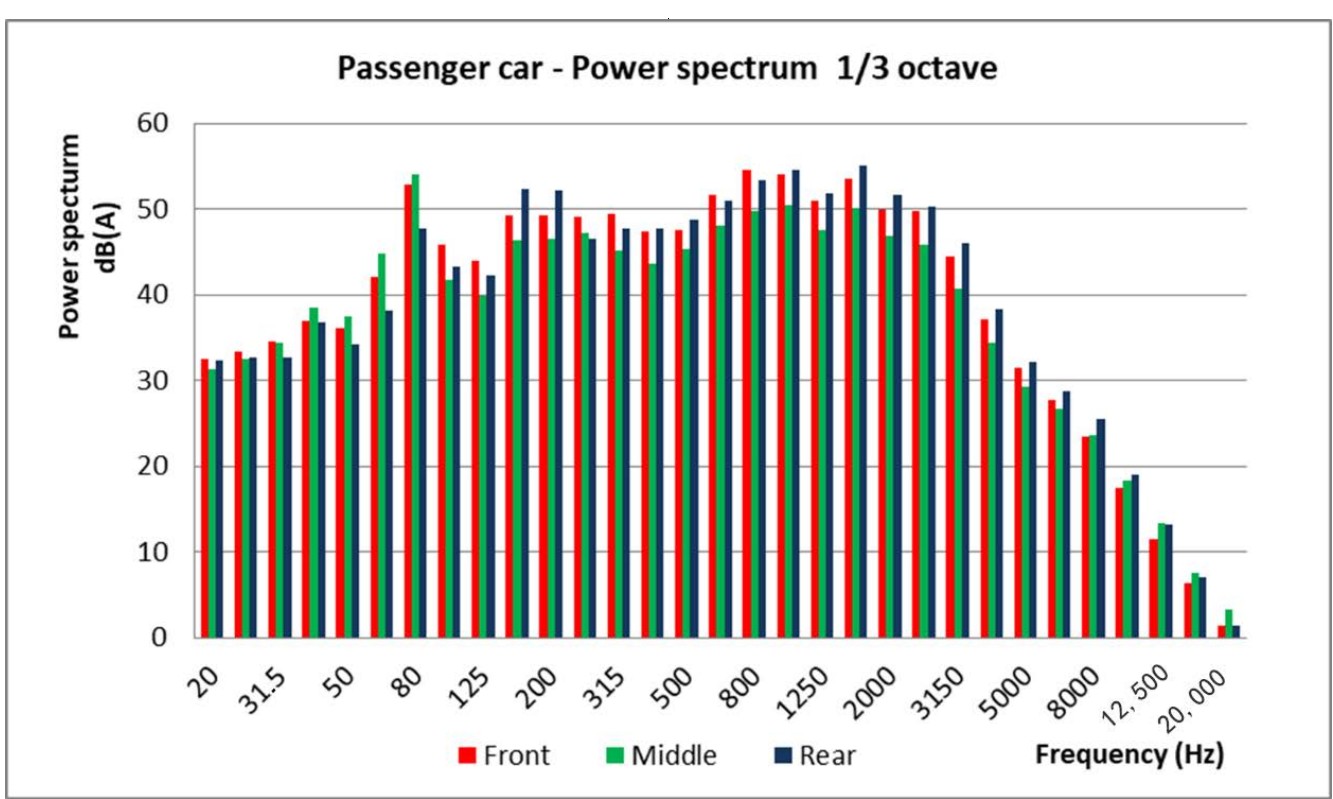

**Figure 8.** 1/3 octave frequency analysis of noise measured in interior of passenger car.

In this case "Online beamforming", in the range of 1500–9300 Hz was used. In online beamforming, it is not possible to average the sound pressure levels due to the fast passage of the wagon. For averaging the fixed position of camera should be used. Here it was not possible to measure the external sound level on the roof of the carriage by a fixed camera.

Figure 9 shows that the front of the passenger coach roof emits noise in the measured frequency range. The other side of the wagon is not visible on the camera due to a decrease in sound pressure level (with respect to the measured position) with a distance above the selected sensitivity range of 2.2 dB. For better visualisation it is necessary to place the camera on the roof of the measured coach and measure it while running, but multiple cameras need to be used within the image. For another type of wagon (Figure 10) at the same speed, the dominant source of noise is from the chassis (which is normal operating condition).

Sound pressure level record of the exterior sound of the passing express train at speed of 160 km/h is presented in the Figure 11. As can be seen from the graph, the noise level of individual wagons is different and is influenced by their design and also by the technical condition of each particular wagon.

Frequency spectrum from the passage of the express train, locomotive, dining car and passenger car is in the Figure 12.

The noise propagates from the chassis through the king-pin and suspension to the car-body of the wagon, causing vibrations of the floor, walls and roof. The higher increase of noise was at operating speed of about 158 km/h when mode shapes of roof occurred. This was caused by excitation from the wheelsets rotation at approx. 923 RPM (wheel diameter of 900 mm).

Measurements of external sound showed a high influence of noise from the bogie for the whole train, where its parts are excited, especially for low frequencies up to 200 Hz. The cooling fans of the locomotive's traction motors cause an increase in noise values at frequencies at 4100 Hz, even when the overall noise of the train set is considered.

**Table 1.** Noise values from the interior of a single-compartment passenger.

| 1/3 Octave [Hz] | Front [dB(A)] | Middle [dB(A)] | Rear [dB(A)] |
|---|---|---|---|
| 20 | 32.5 | 31.3 | 32.3 |
| 25 | 33.3 | 32.5 | 32.6 |
| 31 | 34.6 | 34.3 | 32.7 |
| 40 | 36.9 | 38.5 | 36.8 |
| 50 | 36.1 | 37.5 | 34.2 |
| 63 | 42.1 | 44.8 | 38.2 |
| 80 | 52.9 | 54.0 | 47.7 |
| 100 | 45.8 | 41.7 | 43.2 |
| 125 | 43.9 | 39.8 | 42.3 |
| 160 | 49.3 | 46.3 | 52.3 |
| 200 | 49.3 | 46.6 | 52.1 |
| 250 | 49.1 | 47.2 | 46.5 |
| 315 | 49.5 | 45.2 | 47.7 |
| 400 | 47.3 | 43.6 | 47.8 |
| 500 | 47.5 | 45.4 | 48.8 |
| 630 | 51.7 | 48.0 | 51.0 |
| 800 | 54.6 | 49.8 | 53.4 |
| 1000 | 54.0 | 50.4 | 54.5 |
| 1250 | 50.9 | 47.6 | 51.9 |
| 1600 | 53.5 | 50.1 | 55.0 |
| 2000 | 50.0 | 46.8 | 51.7 |
| 2500 | 49.7 | 45.9 | 50.3 |
| 3150 | 44.5 | 40.7 | 46.0 |
| 4000 | 37.2 | 34.3 | 38.3 |
| 5000 | 31.5 | 29.3 | 32.2 |
| 6300 | 27.7 | 26.7 | 28.7 |
| 8000 | 23.5 | 23.6 | 25.5 |
| 10,000 | 17.4 | 18.4 | 19.0 |
| 12,500 | 11.5 | 13.4 | 13.2 |
| 16,000 | 6.3 | 7.6 | 7.0 |
| 20,000 | 1.4 | 3.2 | 1.4 |

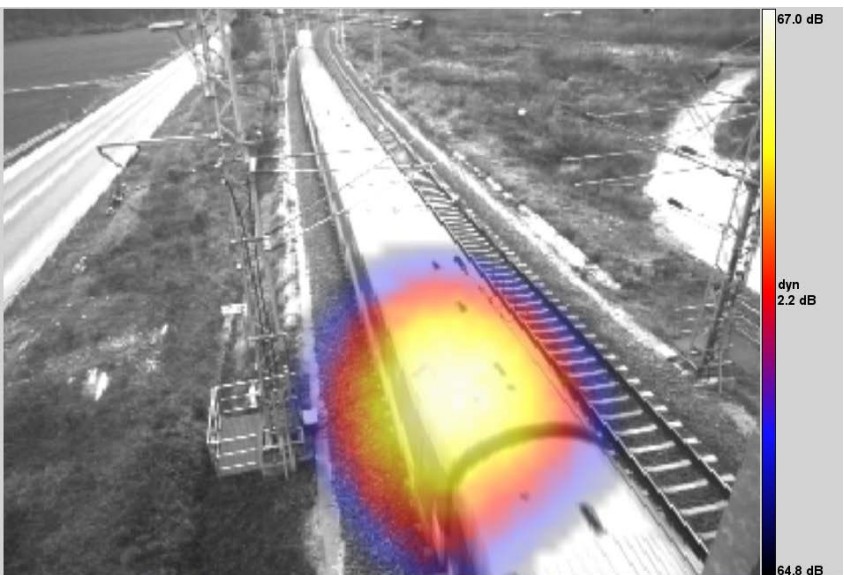

**Figure 9.** Exterior noise map of the passenger coach—end of coach.

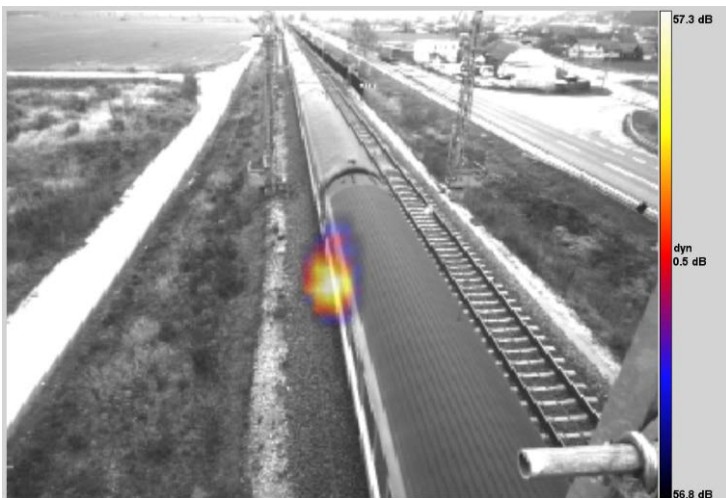

**Figure 10.** Exterior noise map of the passenger coach—bogie.

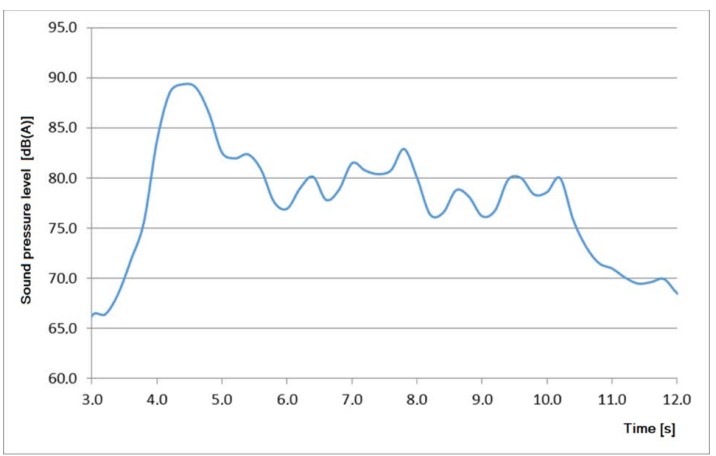

**Figure 11.** Sound pressure level record of the exterior sound—express train at speed of 160 km/h.

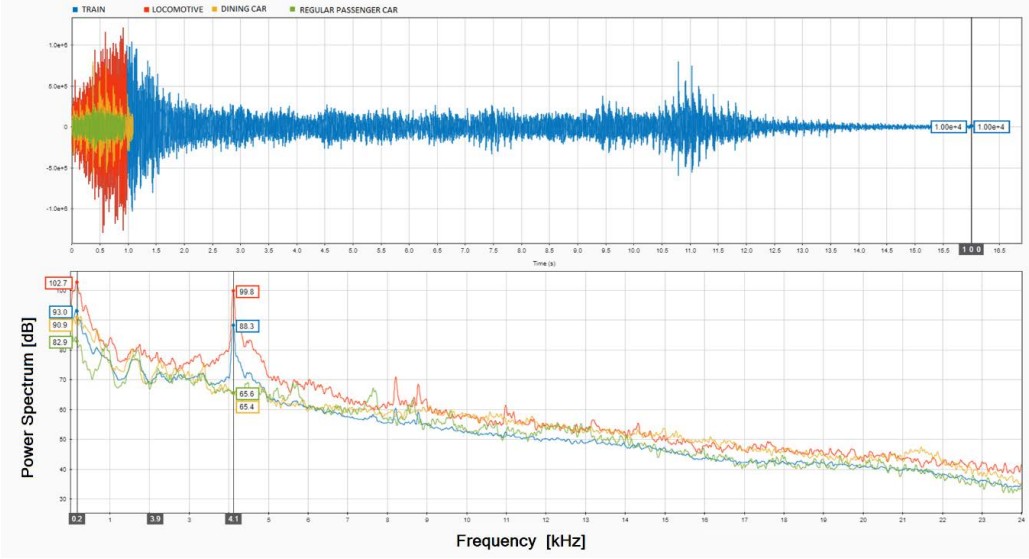

**Figure 12.** Frequency spectrum from the passage of the express train—comparison of locomotive, dining car, regular passenger car and the whole train.

## 4. Sound Insulation Materials Utilization

An innovation in the field of insulation materials is a material made of polyester fibres produced by a technology that lays fibres in three dimension orientation. These insulation materials have advantages over standard mineral insulations mainly in their flexibility, which has a favourable effect on noise absorption and durability, as they are more resistant to vibrations. A significant advantage is the property of higher thermal capacity compared to mineral fibres, which results in lower operating costs, in terms of energy demands for heating and cooling of wagons.

At the same time, this new insulation is an ecological product, as it is 85% made of recycled materials and can be recycled even at the end of its service life. From the point of view of safety, the insulation meets the strictest criteria of European railway standards, namely EN45545—2 + A1 categorization HL3.

However, even with the use of the best noise insulation materials, it is sometimes difficult to achieve the required parameters when the limit factor is the structure of the railway wagon. If the structure of the wagon itself starts to oscillate more significantly in certain running regimens, even more expensive and better insulation will not help. To effectively suppress this source of noise, it is necessary to modify the structure, which can be relatively expensive, even unprofitable compared to the cost of developing a completely new structure.

Since design changes might be quite a lengthy and costly approval process, there are efforts to improve existing designs (structures) with higher quality anti-noise thermal insulation, which should guarantee acoustic comfort for passengers. When modernizing passenger railway wagons, insulation from new materials with better properties and longer service life is used.

### 4.1. Sound Absorption Porerties of New Insulation Materials

When applying modern fibrous porous materials as sound insulation layers in the wall or floor of a railway car, we are particularly interested in the values of the sound absorption coefficient, which is verified experimentally. The most important feature is the frequency dependence of the sound absorption coefficient. In Figure 13 such a dependence of the value of the sound absorption coefficient of two top materials (glass fibres and textile fibres) is shown.

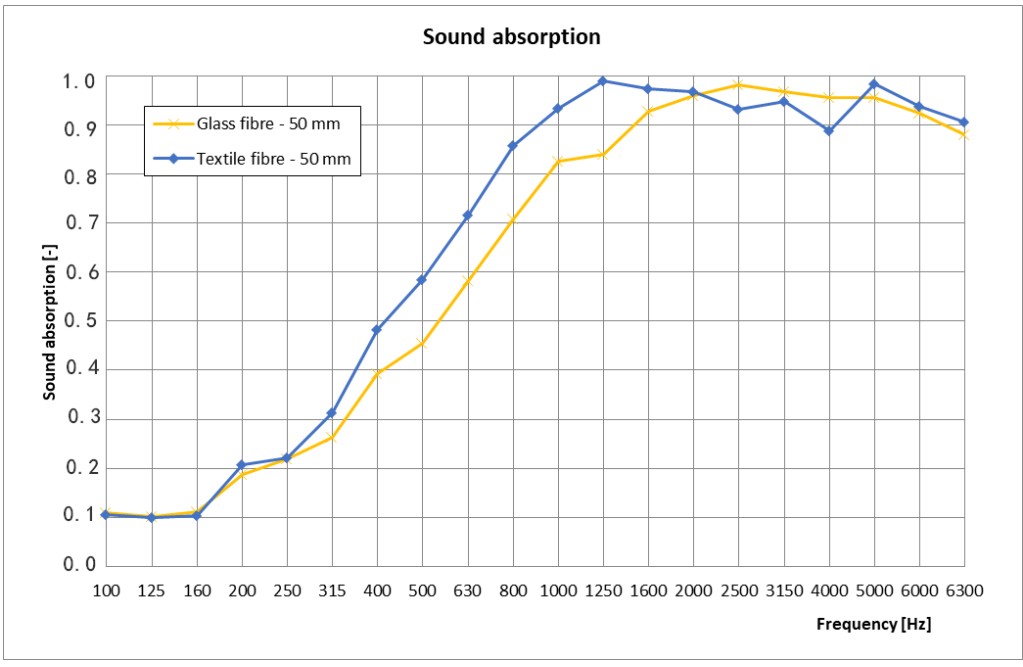

**Figure 13.** Sound absorption coefficient frequency dependences of sound insulation material.

These materials can be used immediately and also in perspective in the walls of railway passenger cars. The experimentally observed values of the absorption capabilities of these materials are significantly higher compared to conventional materials, which have been used for about the last 10 years. At the same time, the mentioned frequency dependences very objectively document the main problem of noise attenuation in railway vehicles, and that is noise attenuation in the frequency range below 160 Hz.

Mineral fibre-based materials are gradually being not used, as their parameters deteriorate significantly with long-term operation. Foam materials based on stricter legislative requirements—EN 45545-2 + A1 do not satisfy the tests requirements of smoke and therefore their application in the interior of railway passenger cars is omitted [34].

### 4.2. Environmental Benefits of the Envizol Train Insulations

Another type of sound insulation material is the Envizol Train. It is a composite insulation material composed of the following layers. The main insulator is composed of polyester fibres in a ratio of 15% of the primary fibre ensuring the connection of fibres. 85% of the fibres are mechanically recycled fibres with a non-combustible treatment that does not contain any hazardous substances. There is a layer of aluminium foil on the surface, which is designed to provide mechanical resistance and has good thermo-reflection parameters, which positively affects the cost of heating the car. There may be a self-adhesive layer on the underframe, which ensures good adhesion and its use saves installation time compared to mechanical anchoring of insulators. This part is not subject to DIN 6701 certification, as it is not a component that affects the safety of railway operation. At the same time, the insulation can also be laminated with a drainage layer, which can drain condensed water to the drain point.

Example of environmental benefits of the Envizol Train insulation is presented in Table 2.

**Table 2.** Envizol Train insulation—characteristic features.

| Characteristics | Quantity |
|---|---|
| Average area of insulation in a passenger car | 250 m$^2$ |
| Average weight of composite insulation in a passenger car | 400 kg |
| Recycled material used in whole composite per passenger car | 300 kg |

The average weight of recycled polyester fibres in a passenger car can be up to 300 kg. For better understanding, if we did not recycle the material and it ended up in a landfill, it would take up 6 cubic meters for 150 years. This means that if there are 15 passenger cars in the train, the environmental saving is one full truck of polyester waste, which has found full use for at least 20 years.

Another case for a better demonstration of the idea of how much recycled material can be used in the insulation of one wagon—1000 pieces of sports functional clothing after the end of its life made of mechanically recycled polyester can be used in the production of insulation for one wagon.

An important aspect is the water consumption for the production of primary fibres. While in the production of primary PES fibre the consumption is on average 51 m$^3$ of water per 1 tonne of fibre, in recycling it is on average 1–2 m$^3$ per tonne of fibre, which leaves a significantly lower footprint and saves natural resources.

## 5. Discussion

In the search for opportunities, risks, possibilities and procedures for reducing the noise of railway vehicles, computer simulation tools and methods of experimental analysis of technical acoustics are particularly strongly interconnected. This leads to the objectification of noise phenomena and events, the visualization of sources and paths of noise propagation, the definition of design and technological procedures for the production as well as operation and maintenance of these vehicles.

Our goal was to present the targeted use of an acoustic camera, in conjunction with a modular sound level meter and FFT analyser to identify external and internal noise of passenger wagons with the aim of improve the ride comfort of passengers. We briefly summarize the results obtained:

- Principal internal noise sources of passenger wagons under the run on real track in operation were detected and visualized (passenger coach roof at operating speed approx. 158 km/h).
- Noise parameters (instantaneous sound pressure levels, equivalent sound pressure levels, frequency spectra) of passenger cars in real operation were measured.
- The noise radiation and noise maps of railway wagons were visualized.
- The properties of sound-insulating materials of classical (glass fibre) and unconventional compositions (recycled textile fibre) were investigated.
- The locations of the highest noise emission of railway passenger wagons were monitored and visualized by sound camera.
- Inputs for optimizing the design of passenger coaches at typical running speed of 160 km/h on Slovak railways form the point of acoustic comfort were defined (mode shapes of roof at speed of 157.9 km/h).

## 6. Conclusions

The results of the experiments confirmed that it is possible to relatively accurately determine the main noise sources by measurements using advanced measuring instrumentation and determining mode shapes of the structure by simulation, and thus propose design countermeasures so that the structure has better acoustic properties. It is known that at the time when the structures of passenger cars, which are currently still in operation, were designed, these tools and possibilities were not available. Such passenger cars have also recently undergone upgrades to improve comfort, using new insulating materials. However, the structure itself, which was designed several decades ago, has its limitations, and if increase the overall comfort (not just acoustic) is expected, further major modifications or a completely new generation of wagon design is likely to be needed.

Sound insulation materials should be used in locations where the noise and vibrations are highest. Besides contribution to sound reduction and higher ride comfort (travel environment in case of passenger wagons), new-generation materials are manufactured from waste. This is secondary benefit to the environment as it reduces the need for new materials and improves environmental friendliness of the railway transport and contributes to its sustainability.

Further research and for more detailed monitoring of the technical condition of railway vehicles by means of acoustic diagnostics is necessary for further progress in silent railway in Slovakia. For example, better visualization of low sound frequencies is possible only by use of an analogue acoustic camera with special microphones for low frequencies (range from 16 Hz) is necessary. Portion of low frequencies is substantial in noise spectra of railway wagon (as shown in Figure 12).

**Author Contributions:** Data curation, L.L. and J.K.; Investigation, J.Ď.; Methodology, J.Ď., P.Z. and J.G.; Project administration, J.Ď.; Validation, P.Z.; Visualization, L.L.; Writing—original draft, J.G.; Writing—review & editing, J.G. All authors have read and agreed to the published version of the manuscript.

**Funding:** This paper is the result of the Project implementation: Competency Centre for Knowledge technologies applied in Innovation of Production Systems in Industry and Services, ITMS: 26220220155, supported by the Research & Development Operational Programme funded by the ERDF. It was also supported by the Scientific Grant Agency of the Ministry of Education of the Slovak Republic and the Slovak Academy of Sciences in project KEGA, no. 018ŽU-4/2018: "Innovation of didactic approaches and content of subjects of technical diagnostic as a tool for enhancing the quality of professional knowledge for practice needs" and KEGA, no. 044ŽU-4/2019: Implementation of innovative elements in the education process within the study program Maintenance of means of transport".

**Institutional Review Board Statement:** Not applicable.

**Informed Consent Statement:** Not applicable.

**Data Availability Statement:** The data presented in this study are openly available by authors.

**Acknowledgments:** The authors would like to express their appreciations to the anonymous reviewers and the editors.

**Conflicts of Interest:** The authors declare no conflict of interest.

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
