# Peer review of "Localization of Increased Noise at Operating Speed of a Passenger Wagon"

_sustainability, doi:10.3390/su13020453_

Round 1

Reviewer 1 Report

The article is well structed and detailed.

Author Response

Thanks for a positive review

Reviewer 2 Report

Please, see document attached

Author Response

Dear reviewer,

Thank you for your comments in the review of our article.
We see them as valuable inputs to increase the overall quality of the article and also as a motivation for our further research in the field of reducing noise on the railway and improving the environmental parameters of railway vehicles.
We incorporated your comments into the text and figures as was suggested.

The responses and the modified article you can find in attachments.

As the English quality concerns - when the content is approved we will use the proof writing offered by the publisher as we are not native English speakers.

Best regards and best wishes for a Happy and covid-free New Year 2021,

Juraj Grencik

Reviewer 3 Report

The submitted paper fits into the very important subject of railway noise. While the argument is actual and needing scientists’ attentions, the paper needs to be re-edited as it is not properly written for a scientific paper. Moreover, further explanations and descriptions are needed in order to explain the methodology.

At first, it appears evident that authors are non-English native speakers. Please improve the writing.

In the abstract, sentence “Although the railway transport is considered to be environmental friendly” is misleading. Without references, I believe it is referred to CO2 emission if compared to road transportation. Said so, in the introduction the related sentences should be better exploited and referenced, while in the abstract it should be removed.

Introduction should be improved with many more references in the sector in order to show how important is the topic. Example are very important papers not named (Licitra, Gaetano, et al. "Annoyance evaluation due to overall railway noise and vibration in Pisa urban areas." Science of the total environment 568 (2016): 1315-1325; Talotte, C., et al. "Identification, modelling and reduction potential of railway noise sources: a critical survey." Journal of Sound and Vibration 267.3 (2003): 447-468; Bunn, Fernando, and Paulo Henrique Trombetta Zannin. "Assessment of railway noise in an urban setting." Applied acoustics 104 (2016): 16-23.) but also, very recent one (Smith, Michael G., et al. "Effects of ground-borne noise from railway tunnels on sleep: A polysomnographic study." Building and Environment 149 (2019): 288-296; Elmenhorst, Eva-Maria, et al. "Comparing the Effects of Road, Railway, and Aircraft Noise on Sleep: Exposure–Response Relationships from Pooled Data of Three Laboratory Studies." International journal of environmental research and public health 16.6 (2019): 1073.). In fact, a good paper on railways noise should include these, as well as a good references section.

The end of the introduction should introduce what the paper will do, as now it is not reported or it is not clear at all.

Please report more details about the acoustic camera used. Most important is its diameter and frequency range. Without these information, Figure 5 has no meaning. What frequencies does it shows?

Figure 6: units are missing. If they are dB, please avoid double decimals in acoustics.

Table 1: no need to report ,0 after the frequencies.

In English writing, use . and not , as decimals.

Also conclusions should be improved by summarizing the paper methodology, why and how you do that.

Author Response

(The authors gave the same response as above.)

Round 2

Reviewer 2 Report

The authors have tried to slightly modify the original manuscript making some minor modifications to the text, but they have not considered at all the main requests raised by the reviewer. At the end, the manuscript seems more like a technical report than a research paper, and I do not think that it accomplish with the level for the journal.

Author Response

Thanks for the review - we tried to make corrections according to your comments.

Reviewer 3 Report

The authors followed my suggestions and the paper is ready for being published. 
As reported by the authors themselves, the english writing should be improved.

Author Response

Thanks for the review - we appreciate you were satisfied with our corrections according to your comments and recommendations.